# Can I Leave the Past Behind? Associations between Childhood Abuse and Adult Psychopathology

**DOI:** 10.3390/healthcare12030412

**Published:** 2024-02-05

**Authors:** Maayan Nagar, Ora Nakash

**Affiliations:** 1Department of Criminology, Ariel University, Ariel 4070000, Israel; 2School for Social Work, Smith College, Northampton, MA 01063, USA; onakash@smith.edu

**Keywords:** childhood abuse, trauma, assessment, borderline personality disorder, post-traumatic stress disorder, mood and anxiety disorders

## Abstract

Background: Research consistently shows that abuse during childhood is related to adult psychopathology. Information regarding childhood abuse is frequently collected from either previous documentation or from participants’ self-reports. Objective: In the current study, we combined information on reports of childhood abuse from several informants (patients, treating clinicians, and independent interviewer), as well as diagnostic assessments of adult patients based on independent interviewer assessments based on structured diagnostic interviews (SCID) and clinician judgments, to better examine the association between exposure to abuse during childhood and adult psychopathology. Participants and Setting: A convenience sample of patients in community mental health and hospital-based clinics (N = 170) and their clinicians (N = 80) participated in the study. Methods: Patients and clinicians completed the Clinical Data Form. Patients also completed the Childhood Trauma Questionnaire. Independent interviewer-assessed patients and completed the Familial Experiences Interview. Clinicians completed a diagnostic assessment of their patients based on clinical judgment. Independent interviewers completed the Structured Clinical Interview (SCID). Results: Reports of exposure to physical, emotional, and sexual abuse during childhood from all informants correlated with the treating clinician’s diagnosis of borderline personality disorder (BPD; r range 0.23–0.37, *p* < 0.05), but not with SCID diagnosis of BPD. Clinician and SCID diagnosis of post-traumatic stress disorder (PTSD) correlated with patient and interviewer reports of childhood sexual abuse (r range 0.23–0.30, *p* < 0.05), but there were no correlations with treating clinician’s reports of sexual abuse. Conclusions: The association between BPD and childhood abuse is consistent across different ratings and measurements. The study raises questions of the ability of a structured interview to accurately capture BPD and highlights the connection between sexual abuse and PTSD, and the importance of treating clinicians’ examination of childhood sexual abuse among their patients.

## 1. Introduction

Adverse childhood experiences, including emotional, physical, and sexual abuse, have been consistently linked to mental health problems in children and adults [1,2]. A review of 52 studies on in-patient care [3] found that more than 50% of the men and over 60% of the women were exposed to sexual or physical abuse in their childhood. Rates of reported childhood abuse in clinical populations vary for different types of abuse, with studies showing the highest prevalence for emotional abuse (25–54%) [4,5], followed by physical abuse (21–30%) [6] and sexual abuse (4.3–28%) [4,7].

Studies have consistently documented the association between childhood abuse and mental health problems in adulthood, including depression, anxiety, post-traumatic stress disorder (PTSD), substance abuse, eating disorders, personality disorders, and dissociative disorders [8,9]. Additionally, psychiatric patients who have been exposed to sexual or physical childhood abuse were found to have an earlier first admission, longer hospitalizations, received more medication, a higher likelihood to commit suicide and self-mutilate, and a higher symptomatology [10,11]. Maladaptive processes of the emotional-processing system and general physiological reactivity have been assumed to contribute the vulnerability of deleterious sequelae of exposure to childhood traumatic experiences [12,13].

Despite this robust body of literature, the information regarding childhood abuse in research to date has frequently relied on a sole source such as previous documentation (e.g., social services charts) or participants’ self-reports.

### 1.1. Childhood Abuse and Adult Personality Disorders

Ample research has examined the association between exposure to neglect and abuse during childhood and adult diagnoses of personality disorders (PDs) [14,15]. Research most frequently based the investigation on a sample of patients in community mental health clinics or a representative community sample [16]. Studies that have included patients in mental health clinics frequently relied on a sole source to collect information on childhood abuse, including patients’ self-reports and interviews or clinical records [17,18]. Findings show that childhood adverse experiences are particularly prevalent among persons diagnosed with PDs, specifically, borderline PD (BPD), and they are associated with greater symptom severity, as well as with worse psychopathology [19,20]. For example, a study conducted among adults seeking mental health care diagnosed with PDs, relying on self-reported history of abuse, showed that individuals diagnosed with PDs reported increased rates of childhood abuse (73%) and neglect (82%). BPD was, in particular, marked as being associated with childhood abuse and neglect [19]. A study among patients from different mental health care settings (outpatient, inpatient, and forensic) documented associations between specific types of childhood maltreatment and adult PDs. Sexual abuse was associated with symptoms of borderline, paranoid, schizoid, and avoidant PDs; physical abuse was associated with antisocial PD; emotional abuse was associated with paranoid, borderline, and schizotypal PDs; and neglect was associated with borderline and histrionic PDs [20]. Importantly, studies that used multiple sources of information pointed to potential gaps in reports of childhood experiences of abuse. For example, a study by Rossiter et al. [6] among adult patients in community and hospital-based clinics compared information from self-reports and clinical notes. Authors documented that 93–100% of patients with antisocial, passive-aggressive, paranoid, and borderline PDs self-reported childhood-abuse experiences, while only 40–69% of clinical notes included similar information for these patients [6].

Studies that have examined PD symptoms and childhood adversities in the general population (non-clinical samples) have used mainly structured interviews for assessing psychiatric diagnosis (SCID I and II) and self-reports for collecting information on histories of trauma. These assessments have shown similar results. For example, Johnson et al. [21], using psychosocial and psychiatric interviews, found that an exposure to abuse and neglect during childhood was associated with greater PD symptomatology, and that persons that were abused or neglected during childhood were up to four times more likely to be diagnosed with PDs in adulthood compared to those who did not experience maltreatment in childhood. Another study, conducted by Tyrka et al. [22], applying methods of psychiatric diagnostic interviews and self-report questionnaires assessing exposure to abuse during childhood have shown similar results. Childhood maltreatment, including emotional, physical, and sexual abuse, as well as neglect, was associated with elevated symptoms of PDs, specifically of borderline, paranoid, avoidant, obsessive-compulsive, and dependent PDs. A different study also used structured diagnostic interviews, as well as self-reports, to document that childhood adversities, including emotional, physical, and sexual abuse, were highly prevalent among persons with PDs [23]. Indeed, a recent review of 44 studies regarding early-life stress has confirmed the robust connection between emotional abuse and neglect with PDs [24].

### 1.2. Childhood Abuse and Mood and Anxiety Disorders

Similar to the literature on adult PDs and childhood abuse, research has documented the association between exposure to abuse during childhood and symptoms of mood and anxiety disorders in adulthood. A meta-analysis, conducted by Mandelli et al. [25], documented that, regardless of research method, childhood experiences of emotional neglect were highly correlated with adult symptoms of depression [25]. Another meta-analysis showed similar results, indicating that childhood exposure to physical and sexual abuse, as well as neglect, was associated with common adult mental disorders [24]. It is noteworthy that all studies to date have used diagnostic information and information regarding childhood abuse from either one (participants or interviewers) or two sources (participants and independent interviewers), with diagnostic assessment mostly based on structured clinical interviews (e.g., SCID) or self-report questionnaires, and childhood abuse information gathered mostly through self-reports or semi-structured interviews [25,26,27].

Using data from self-report measures, Kounou et al. [26] found that compared to individuals without psychiatric history, patients with major depression reported higher rates of childhood maltreatment, including emotional, physical, and sexual abuse, as well as neglect. Similarly, Mall et al. [28] showed that childhood exposure to emotional abuse and/or neglect predicted adult depression during the preceding year. As for studies using interviews, Hovens et al. [29] reported that childhood exposure to emotional and physical abuse, as well as emotional neglect, was correlated with adult depressive and anxiety disorders. Surprisingly, exposure to childhood sexual abuse was not associated with depression during adulthood. Another study relying on interviews showed that sexual abuse during childhood was related to lifetime mood and anxiety disorders, while physical abuse during childhood was associated with lifetime anxiety disorders [30].

### 1.3. The Current Study

Previous studies have consistently documented the relationship between exposure to abuse during childhood and adult psychopathology. However, previous studies have used information collected mainly from structured interviews and/or self-reports and focused on psychiatric illness rather than on adaptive functioning. The aim of the current study was to examine the association between childhood experiences of abuse (emotional, physical, and sexual) and adaptive functioning, diagnoses of PDs and diagnoses of mood and anxiety disorders among patients in community mental health and hospital-based clinics using a multimethod design relying on multiple informants. For this purpose, we have triangulated information on childhood abuse from three sources (patients, treating clinicians, and independent interviewer), as well as information on psychiatric diagnoses based on structured diagnostic interviews (diagnostic gold standard) and treating clinician’s DSM-IV diagnosis based in clinical judgment.

## 2. Materials and Methods

### 2.1. Study Design and Participants

A convenience sample of 80 clinicians and 170 patients was recruited for this study from eight community mental health and hospital-based clinics providing services to ethnically and socio-economically diverse adult populations in Israel. Patient-inclusion criteria: adult clinical outpatients, over 18 years of age, known reasonably well by their treating clinicians (having between five hours minimum to one year maximum of clinical contact with their treating clinician; *M*_months in treatment_ = 6.0, *SD* = 4.6). We have chosen to include patients who are reasonably well-known to their clinicians in order to make sure clinicians are able to complete the clinical information assessed in this study [31]. Patients’ exclusion criteria: not actively suicidal and psychotic. Clinicians’ inclusion criteria: all active clinicians providing mental health services were invited. No exclusion criteria were applied for clinician participation. The patient sample included 69 adult (*M* = 37.5, ages ranged from 18–70) men (40.6%) and 101 women (59.4%). Eighty clinicians from different disciplines (48 psychologists, 25 social workers, and seven psychiatrists) participated in the study (Table 1 presents complete sample descriptive information).

### 2.2. Procedure

Clinician participants were recruited through informational meetings at each clinic. They were asked to participate with up to four patients each (*M* = 2.13, *SD* = 1.27). Clinicians recommended patients deemed appropriate for the study. Interested patients were contacted by a research coordinator that provided a thorough description of study procedures. Institutional ethics committees at each participating clinic approved all study components, and data collection adhered to acknowledged human subject protocols. Prior to study participation, participating patients and clinicians provided written informed consent.

Each patient participated in two separate 1.5 h sessions: the first session included completion of demographic, diagnostic, and clinical questionnaires. In addition, participants were assessed by an independent evaluator using the Familial Experiences Interview (FEI; [32]). In the second session, the Structured Clinical Interview for DSM-IV Axis I and Axis II Disorders (SCID-I/P and SOCD-II) was administered by an independent licensed clinician. Treating clinicians completed measures evaluating their patients’ clinical information and symptomatology. Both patients and clinicians received USD 50 compensation for their participation. A full description of measures administered in the present study is summarized below. For further details about the study, see Nakash et al. [33].

### 2.3. Measures

#### 2.3.1. Structured Clinical Interview for DSM-IV Axis I and Axis II Disorders (SCID-I/P and SOCD-II)

The SCID [34,35] is a structured clinical interview employed in studies of psychiatric populations to evaluate for DSM-IV Axis I and Axis II disorders. The SCID procedure has been widely used in research and has demonstrated robust reliability and validity [36,37]. Four independent, experienced research interviewers (licensed clinical psychologists) administered the SCID. A dimensional score was produced for each PD diagnosis based on the number of criteria participants met for each PD SCID diagnosis.

#### 2.3.2. Categorical DSM-IV Diagnosis [38]

Clinicians determined the presence or absence of personality disorder diagnoses based on their clinical judgment. To guarantee that DSM-IV criteria were used, clinicians were provided with criteria checklists for all personality disorders relevant for the study. Respondents were required to discern whether each diagnostic feature was present or absent and establish the presence/absence of the disorder based on these conclusions.

#### 2.3.3. Clinical Data Form (CDF)

The CDF is a questionnaire that indexes demographic, diagnostic, and etiological information. It is attainable as clinician-report, patient-report, and interview-based forms. The CDF has been used extensively in research [39,40]. In this study, the CDF was completed by the SCID independent interviewer and the patient’s treating clinician. The patient filled in a self-report version of the measure. All raters in the present study examined the patient’s overall psychological functioning (1 = severe personality problems; 5 = high functioning), quality of social and romantic relationships (ranged 1 = unstable/absent/conflictual; 5 = stable/strong/loving), social support (number of close confidants: 1 = none, 2 = very few, 3 = some, 4 = many); physical health (1 = serious/degenerative illness; 5 = few/occasional health concerns), and patient’s clinical history, including history of suicidal and self-mutilating behavior, psychiatric hospitalization, legal trouble necessitating arrest within the past 5 years, job loss resulting from interpersonal conflicts within the past 5 years, and childhood and adulthood physical abuse (all items rated as “yes” or “no”). For childhood abuse specifically, the following language was used: “Was the patient physically abused as a child?” and “Was the patient sexually abused as a child?” Response options were “no”, “unclear”, and “yes”. Clinicians and interviewers also completed the Global Assessment of Functioning (GAF) [38], a scale used by skilled clinicians to rate patient’s psychosocial functioning [41].

Based on the CDF and GAF, we created a Global Composite Assessment of Adaptive Functioning (GCAAF) that stipulates a comprehensive measure of adaptive functioning across all significant life domains [40]. The scale was computed as mean score of Global Assessment of Functioning (GAF), overall personality functioning, and items inspecting the state of interpersonal and educational/occupational functioning and support. Previous research has established high reliability of adaptive functioning ratings and their notable correlation with ratings determined by independent interviewers [42,43]. Clinician reports on the CDF meaningfully correlate with a patient-report version of the instrument, with leading diagnostic potency [40,44].

Global Composite Assessment of Adaptive Functioning measures for each informant were created by standardizing (Z-scoring) and then averaging their ratings across various items, assessing a particular dimension and then accumulating the standardized items across all three raters (the research interviewer, the treating clinician, and patient self-report) produced indications of adaptive functioning from numerous viewpoints [31]. Compound scale internal consistency (Cronbach’s α) = 0.86; ICC across the three raters = 0.87.

#### 2.3.4. Childhood Trauma Questionnaire (CTQ; [45])

Participating patients recorded responses to the CTQ, a 28-item self-report scale surveying five dimensions of adverse childhood experiences: emotional neglect, emotional abuse, physical neglect, physical abuse, and sexual abuse. Items are ranked along a five-point Likert scale (1 = never true; 5 = very often true). Summarized subscale scores were computed to denote abuse severity. The present study employed the following subscales: emotional, physical, and sexual abuse. All subscales demonstrated excellent internal consistency (ranging from α = 0.73 to 0.92).

#### 2.3.5. The Familial Experiences Interview (FEI; [32])

The Familial Experiences Interview (FEI; [32]) assesses physical and sexual abuse, physical neglect, loss, and other family experiences. The FEI has displayed strong psychometric properties in studies of BPD populations [46]. FEI scores were determined by the presence or absence of emotional, physical, or sexual abuse. The number of experiences for each type of abuse were calculated to designate severity of abuse.

### 2.4. Data Analysis

We computed Chi-square tests to assess the associations between categorical variables and Cramer’s V and Phi to assess the strength of correlations. We computed Pearson’s r to assess association between continuous variables and Kendall’s Tau for ordinal variables. We used Benjamini and Hochberg [47] False Discovery Rate (FDR) to control for multiple comparisons.

## 3. Results

Table 2, Table 3 and Table 4 summarize the results for correlations between physical, emotional, and sexual abuse that were reported by the different informants (patients, treating clinicians, and independent interviewer) with SCID and treating clinicians’ diagnosis of any PD, BPD, mood and anxiety disorders, and PTSD. Results show significant negative moderate correlations between patients’ and interviewers’ reports of adult adaptive functioning with childhood physical and emotional abuse (ranging from −0.23 to −0.28). There were no significant associations between treating clinician’s reports of adult adaptive functioning and childhood physical abuse.

Treating clinicians’ diagnoses of BPD correlated significantly with reports of emotional, physical, and sexual childhood abuse made by all three informants (patient clinician and interviewer) (ranging from 0.23 to 0.31), regardless of the type of measure used. In contrast, the SCID diagnosis of BPD correlated solely with emotional abuse as reported by the interviewer. The diagnosis of any PD followed a similar trend. Treating clinicians’ diagnoses of any PD correlated significantly with the patients’ reports of exposure to emotional, physical, and sexual abuse during childhood (ranging from 0.23 to 0.29), as well as with clinicians’ reports of sexual abuse and interviewers’ reports of emotional abuse. The SCID diagnosis of any PD correlated significantly only with clinicians’ reports of physical abuse and interviewers’ reports of emotional abuse.

As expected, both treating clinicians’ and SCID diagnosis of PTSD correlated significantly with patients’ and interviewers’ reports of sexual abuse (ranging from 0.23 to 0.30). The SCID lifetime diagnosis of PTSD correlated with interviewers’ reports of physical abuse, in addition to patients’ and interviewers’ reports of sexual abuse. Any current mood or anxiety disorder, either diagnosed by the treating clinician (using clinical judgment) or the interviewer (using the SCID), was not related to any report of childhood abuse. Any lifetime mood or anxiety disorder diagnosed by the interviewer based on the SCID was correlated with patients’ and interviewers’ reports of sexual abuse.

## 4. Discussion

The current study is the first to explore the relationship between exposure to abuse during childhood and adult psychopathology (i.e., PDs, mood and anxiety disorders) while triangulating clinical information from multiple sources (patients’ reports, treating clinician assessments, and independent interviewer evaluations). Our main findings show that reports of childhood emotional, physical, and sexual abuse across informants correlated with treating clinicians’ diagnoses of BPD. This finding supports the results of past studies emphasizing the increased prevalence of exposure to abuse during childhood among people diagnosed with BPD in adulthood [6,19]. The directionality of the impact of exposure to childhood abuse on the development of BPD is a source of debate through decades of research. Recent reviews and meta-analyses support the notion that exposure to abuse during childhood may lead to the development of BPD in adulthood [48,49]. Our study shows that the association between exposure to abuse during childhood and adult BPD persists across different reporters and assessment instruments used.

Interestingly, most reports of childhood abuse in our study did not correlate with the SCID BPD diagnosis. Our results regarding correlations of any PD diagnosis and childhood abuse follow a similar trend. It is unclear whether these differences derive from treating clinicians’ misdiagnoses or from problems with the validity of SCID interviews to assess PDs. Previous criticisms of structured psychiatric interviews point out the difficulty in establishing a valid PD diagnosis. One criticism focuses on the fact that structured psychiatric interviews rely on the patient’s self-reported symptomatology. According to Westen and Shedler [50], this is problematic due to several reasons. First, there are facets of personality pathology that one may not be aware of in him/herself. Second, a large portion of human behavior reflects psychological processes that are consciously unreportable. Another criticism is that since PDs focus on patterns of interpersonal functioning, their assessment requires clinical judgment that is based on clinical experience, as opposed to other DSM-based diagnoses (such as mood and anxiety) that have more specific symptomatic criteria [51]. Since all PDs are characterized by disturbances in adaptive functioning, the fact that our study also documented an association between childhood abuse and adaptive functioning provides support and validation to the clinician’s BPD diagnosis. Yet, additional studies are required to examine this controversy.

Our findings further document the correlation between PTSD diagnosis and reports of childhood sexual abuse. In this case, similar patterns of correlations were found with both treating clinician and PTSD diagnoses based on SCID. Interestingly, while a PTSD diagnosis was associated with patients’ and interviewers’ reports of sexual abuse, there were no correlations with treating clinicians’ reports of sexual abuse. In a previous study, we have shown that treating clinicians report lower rates of childhood sexual abuse as compared with their adult patients’ self-reports [52]. Thus, it remains unclear whether treating clinicians do not collect sufficient information regarding sexual abuse, or if they do not relate these symptoms with a PTSD diagnosis [53,54,55].

Unexpectedly, our findings did not show significant associations between exposure to abuse during childhood and any mood and anxiety disorders in adulthood across all informants. Results from previous studies are inconsistent, with some showing correlations between specific types of childhood abuse and lifetime mood and anxiety disorders [30]; others show correlations with current mood and anxiety disorders [28], and others show no significant correlation between exposure to abuse during childhood adult mood and anxiety disorders [56]. The most consistent finding in past research regarding childhood adverse experiences and mood and anxiety disorders relates to emotional and physical neglect. Unfortunately, we did not include an assessment of these variables in the current study, and future studies are warranted to examine this connection across informants.

Our study has limitations. First, the data were collected before the publication of the DSM 5. However, since the categorical basis for the diagnostic system has not changed in this recent version, nor has the core criteria for major mood and major anxiety disorders changed for the diagnosis of personality disorders, we believe that our findings can be extrapolated to the current version. The generalizability of the findings may be limited as we included a convenience sampling method, which may have been subject to selection bias. In addition, clinicians’ identification of their patients’ histories of abuse was based on two questions assessing sexual and physical abuse, and we were not able to collect information regarding clinicians’ identification of exposure to emotional abuse and neglect. Due to the large number of variables tested in this manuscript and the relatively small number of participants, we were unable to examine multivariate predictive models or to check for confounding variables. Future studies should elaborate our findings and examine relevant interactions and analyze the role that some variables may play as effect modifiers (such as age, gender, etc.). Finally, the cross-sectional design of the study does not allow causal relationships to be established. Therefore, the conclusions are limited.

Our study provides additional support to the link between exposure to abuse during childhood and BPD diagnosis using data from multiple sources, including the patient, treating clinician, and independent interviewer, as well as the importance of including information from multiple sources when assessing exposure to childhood trauma and its mental health sequela in adulthood.

## 5. Conclusions

Examining complex clinical topics such as the association of exposure to childhood adverse events and adult psychopathology warrants the triangulation of research methods and informants. Using rigorous methodology that utilizes a multi-method and multi-informant design increases the credibility and validity of the findings. This is especially true for the clinical context, where “objective” diagnostic tests do not exist and diagnostic assessment must rely on clinical interviews. Acknowledging the strength and limitations of the different viewpoints of patients, clinicians, and independent observers in offering clinical assessments of patients is critical to avoid bias in diagnoses. When possible, mental health clinicians should collect diagnostic information from multiple informants and utilize self-report measures, structured diagnostic interviews, and clinical judgment when assessing their patients.

## Figures and Tables

**Table 1 healthcare-12-00412-t001:** Sociodemographic and clinical information for patients and clinician participants in the study.

	Patients (n = 170) % (N)	Clinicians (n = 80) % (N)
Gender		
Male	40.6% (69)	25.0% (20)
Female	59.4% (101)	75.0% (60)
Age; *M (SD)*	37.5 (13.7)	41.6 (9.7)
Country of birth		
Israel	75.3% (128)	90.0% (72)
Other	24.1% (41)	8.8% (7)
Relationship status		
Single	43.5% (74)	
With longtime partner, not married	5.3% (9)	
Married	32.9% (56)	
Separated/divorced	15.9% (27)	
Widowed	1.8% (3)	
Education		
Less than 12 years	12.4% (21)	
High school graduate	20.6% (35)	
Certification studies	11.8% (20)	
Some college or technical school	21.8% (37)	
Technical school graduate	4.1% (7)	
College graduate	17.1% (29)	
Some graduate school	4.1% (7)	
Completed graduate school	6.5% (11)	
Currently employed		
No	37.6% (64)	
Yes	60.0% (102)	
Household income ^1^		
Primarily student	10.6% (18)	
0–1300$	31.2% (53)	
1301–2600$	24.1% (41)	
2601–3900$	15.3% (26)	
3901–5200$	2.9% (5)	
More than 5200$	7.7 (13)	
Months in treatment with current clinician; *M (SD)*	6.0 (4.6)	
Number of sessions; *M (SD)*	16.2 (8.4)	
Discipline		
Psychiatry		8.8% (7)
Psychology		60.0% (48)
Social work		31.3% (25)
Level or expertise		
Licensed		71.3% (57)
Trainee		27.5% (22)
Years of experience post-training; *M (SD)*		8.7 (10.9)

Note: Numbers do not add up to 100% due to missing data. ^1^ Mean household income in Israel approximately 4000$ per month.

**Table 2 healthcare-12-00412-t002:** Correlations between childhood experiences of abuse as reported by patients with adaptive functioning, SCID diagnosis, and clinicians’ DSM diagnosis (N = 170).

		SCID Diagnosis					Clinician’s DSM Diagnosis				
	Global Adaptive Functioning	Any PD	BPD	Any Mood or Anxiety Past Month	MDD Past Month	PTSD Past Month	Any PD	BPD	Any Mood or Anxiety	MDD	PTSD
		(n = 72; 42.4%)	(n = 21; 12.4%)	(n = 69; 40.6%)	(n = 34; 20.0%)	(n = 19; 11.2%)	(n = 50; 29.4%)	(n = 19; 11.2%)	(n = 77; 45.3%)	(n = 42; 24.7%)	(n = 20; 11.8%)
Reported by patient											
Physical abuse (CDF)	−0.20 ***	0.22 **	0.21 **	0.08	0.04	0.18 *	0.18 *	0.29 ***	0.06	0.01	0.16 *
Sexual abuse (CDF)	−0.20 **	0.14	0.19 *	0.04	0.09	0.22 **	0.29 ***	0.37 ***	0.13	0.06	0.08
Emotional abuse (CTQ)	−0.28 ***	0.17 *	0.22 ***	0.10	0.05	0.14 *	0.29 ***	0.30 ***	0.08	0.02	0.16 *
Physical abuse (CTQ)	−0.23 ***	0.12	0.16 *	0.06	0.05	0.17 *	0.23 **	0.31 ***	0.04	−0.06	0.14
Sexual abuse (CTQ)	−0.18 **	0.20 **	0.18 *	0.16 *	0.11	0.26 ***	0.22 **	0.29 ***	0.03	−0.08	0.23 ***

* *p* < 0.05, ** *p* < 0.01, *** *p* < 0.001. Strongest colors indicate stronger correlations. Correlations below an absolute value of 0.23 were deemed non-significant using Benjamini and Hochberg’s (1995) FDR pairwise procedure. n indicates the number of participants diagnosed with each criterion. SCID—Structured Clinical Interview for DSM-IV diagnoses; DSM—Diagnostic and Statistical Manual of Mental Disorders; CDF—Clinical Data Form; CTQ—Childhood Trauma Questionnaire; PD—Personality Disorders; MDD—Major Depressive Disorder; PTSD—Post-traumatic Stress Disorders; BPD—Borderline Personality Disorder. Correlations for each of the CDF variables with each SCID diagnosis and each clinician’s DSM diagnosis were calculated with Chi-square tests and Cramer’s V and Phi for strength of correlation. Correlations for each of the CTQ variables with each SCID diagnosis and each clinician’s DSM diagnosis were calculated with Kendall’s Tau. For each CTQ variable and Global Adaptive Functioning, Pearson’s r correlations were calculated.

**Table 3 healthcare-12-00412-t003:** Correlations between childhood experiences of abuse as reported by clinicians with adaptive functioning, SCID diagnosis, and clinician’s DSM diagnosis (N = 170).

		SCID Diagnosis					Clinician’s DSM Diagnosis				
	Global Adaptive Functioning	Any PD	BPD	Any Mood or Anxiety Past Month	MDD Past Month	PTSD Past Month	Any PD	BPD	Any Mood or Anxiety	MDD	PTSD
		(n = 72; 42.4%)	(n = 21; 12.4%)	(n = 69; 40.6%)	(n = 34; 20.0%)	(n = 19; 11.2%)	(n = 50; 29.4%)	(n = 19; 11.2%)	(n = 77; 45.3%)	(n = 42; 24.7%)	(n = 20; 11.8%)
Reported by clinician											
Physical abuse (CDF)	−0.20 ***	0.10	0.20 *	0.03	−0.01	0.05	0.12	0.26 ***	0.04	0.00	0.1
Sexual abuse (CDF)	−0.13	0.08	0.19 *	0.08	0.06	0.13	0.25 **	0.26 ***	0.08	0.08	0.13

* *p* < 0.05, ** *p* < 0.01, *** *p* < 0.001. Strongest colors indicate stronger correlations. Correlations below an absolute value of 0.23 were deemed non-significant using Benjamini and Hochberg’s (1995) FDR pairwise procedure. n indicates the number of participants diagnosed with each criterion. SCID—Structured Clinical Interview for DSM-IV diagnoses; DSM—Diagnostic and Statistical Manual of Mental Disorders; CDF—Clinical Data Form; PD—Personality Disorders; MDD—Major Depressive Disorder; PTSD—Post-traumatic Stress Disorders; BPD—Borderline Personality Disorder. Correlations for each of the CDF variables with each SCID diagnosis and each clinician’s DSM diagnosis were calculated with Chi-square tests and Cramer’s V and Phi for strength of correlation. For each CDF variable and Global Adaptive Functioning, Kandell’s Tau correlations were calculated.

**Table 4 healthcare-12-00412-t004:** Correlations between childhood experiences of abuse as reported by independent interviewer with adaptive functioning, SCID diagnosis, and clinician’s DSM diagnosis (N = 170).

		SCID Diagnosis					Clinician’s DSM Diagnosis				
	Global Adaptive Functioning	Any PD	BPD	Any Mood or Anxiety Past Month	MDD Past Month	PTSD Past Month	Any PD	BPD	Any Mood or Anxiety	MDD	PTSD
		(n = 72; 42.4%)	(n = 21; 12.4%)	(n = 69; 40.6%)	(n = 34; 20.0%)	(n = 19; 11.2%)	(n = 50; 29.4%)	(n = 19; 11.2%)	(n = 77; 45.3%)	(n = 42; 24.7%)	(n = 20; 11.8%)
Reported by interviewer											
Emotional abuse (FEI)											
Number of experiences	−0.25 ***	0.21 **	0.24 ***	0.13	0.09	0.11	0.23 ***	0.26 ***	0.02	−0.01	0.02
At least one	−0.19 **	−0.25 ***	0.15	0.12	0.16	0.10	0.12	0.19 *	−0.04	−0.1	0.01
Sexual experiences (FEI)											
Number of experiences	−0.20 ***	0.16 *	0.19 *	0.18 *	0.19 *	0.26 ***	0.19 **	0.28 ***	0.09	0.02	0.30 ***
At least one	−0.19 **	0.12	0.17 *	0.17 *	0.17 *	0.25 **	0.21 **	0.28 ***	0.08	0.02	0.30 ***
Physical punishment (FEI)											
Number of experiences	−0.25 ***	0.21 **	0.14	0.19 **	0.11	0.19 *	0.16 *	0.26 ***	0.05	−0.04	0.09
At least one	−0.25 ***	0.25 ***	0.12	0.24 **	0.13	0.12	0.18 *	0.23 **	0.03	−0.07	0.06

* *p* < 0.05, ** *p* < 0.01, *** *p* < 0.001. Strongest colors indicate stronger correlations. Correlations below an absolute value of 0.23 were deemed non-significant using Benjamini and Hochberg’s (1995) FDR pairwise procedure. n indicates the number of participants diagnosed with each criterion. SCID—Structured Clinical Interview for DSM-IV diagnoses; DSM—Diagnostic and Statistical Manual of Mental Disorders; FEI—The Familial Experiences Interview; PD—Personality Disorders; MDD—Major Depressive Disorder; PTSD—Post-traumatic Stress Disorders; BPD—Borderline Personality Disorder. Correlations for each of the FEI with at least one variable with each SCID diagnosis and each clinician’s DSM diagnosis were calculated with Chi-square tests and Cramer’s V and Phi for strength of correlation. Correlations for each of the FEI number-of-experience variables with each SCID diagnosis and each clinician’s DSM diagnosis were calculated with Kendall’s Tau. For each FEI number-of-experience variable and Global Adaptive Functioning, Pearson’s r correlations were calculated.

## Data Availability

Data will be made available upon request.

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
