# Peer review of "Can I Leave the Past Behind? Associations between Childhood Abuse and Adult Psychopathology"

_healthcare, 2024, doi:10.3390/healthcare12030412_

Round 1
Reviewer 1 Report (New Reviewer)
Comments and Suggestions for Authors
First and foremost, I would like to express my gratitude for the chance to review your work. It is evident that a significant amount of effort and dedication went into this research, and it was a pleasure to read your manuscript.
Upon thorough examination, I found several strengths in your manuscript that are worth commending:
The manuscript offers an exceptionally thorough and comprehensive review of the existing literature. This not only sets a solid foundation for your study but also provides a valuable resource for readers interested in this research area.
-
I was particularly impressed with the methodological rigor demonstrated throughout the study. The careful design and detailed procedures add significant credibility to your findings.
-
Besides, the clarity with which you've drawn your conclusions is commendable. They are well-supported by your data and provide a succinct summary of your research findings.
- While your manuscript is strong in many areas, I would like to suggest a couple of enhancements that could further improve the quality and impact of your work:
-
- Your introduction provides a broad and informative context for your study, which is commendable. However, the specific details of your study seem to be dispersed throughout the introduction, which might potentially overwhelm or confuse the reader early on.
- I suggest concentrating the finer details of your study exclusively in Section 1.3, "The Current Study". This focused approach can help readers better grasp the unique aspects and objectives of your research without having to sift through the general background information.
- I recommend adding a dedicated table or chart that explicitly details the inter-judge agreement for each item assessed by the clinician and interviewer for each dimension evaluated.
- Providing a visual representation of these statistics would not only clarify the extent of agreement between raters but also reinforce the reliability of your data and findings.
- These modifications, I believe, will greatly enhance the readability and scholarly rigor of your manuscript.
-
Author Response
Comment 1
First and foremost, I would like to express my gratitude for the chance to review your work. It is evident that a significant amount of effort and dedication went into this research, and it was a pleasure to read your manuscript. Upon thorough examination, I found several strengths in your manuscript that are worth commending: The manuscript offers an exceptionally thorough and comprehensive review of the existing literature. This not only sets a solid foundation for your study but also provides a valuable resource for readers interested in this research area. I was particularly impressed with the methodological rigor demonstrated throughout the study. The careful design and detailed procedures add significant credibility to your findings. Besides, the clarity with which you've drawn your conclusions is commendable. They are well-supported by your data and provide a succinct summary of your research findings.
Response
Thank you for the positive feedback.
Comment 2
Your introduction provides a broad and informative context for your study, which is commendable. However, the specific details of your study seem to be dispersed throughout the introduction, which might potentially overwhelm or confuse the reader early on. I suggest concentrating the finer details of your study exclusively in Section 1.3, "The Current Study". This focused approach can help readers better grasp the unique aspects and objectives of your research without having to sift through the general background information.
Response
Thank you for this suggestion. We have reorganized the Itroduction following this recommendations and have concentrated all the information about the current study in Section 1.3.
Comment 3
I recommend adding a dedicated table or chart that explicitly details the inter-judge agreement for each item assessed by the clinician and interviewer for each dimension evaluated. Providing a visual representation of these statistics would not only clarify the extent of agreement between raters but also reinforce the reliability of your data and findings.
Response
Thank you for this suggestion. However, since there was no aggregation of interrater data used in this study, it is not possible to add a table for the interrater reliability. Additionally, we have previously reported the inter-rater reliability regarding experiences of abuse derived from this project (see Nagar, Nakash and Westen, 2020). We have shown that all reporters – client, clinician, and interviewer- reported similar information regarding types of abuse experienced by clients.
Reviewer 2 Report (New Reviewer)
Comments and Suggestions for Authors
Dear, the proposed manuscript is well structured in all its parts, effective and clear in its representation, the references are relevant, and there is linearity between findings, discussions, and conclusions. However, presents 3 major critical issues that make me lean toward its rejection. To offer you an opportunity to remedy the structural deficiencies and the appreciable work done, I have decided to choose the major revision, for the reasons I indicate here and hope you can correct:
1) the entire study refers to version IV of the DSM (2000), including the indication of axis subdivisions, when in fact that version has been superseded and today we use the rectified version V (5-TR, 2022). This structural deficiency invalidates the entire editorial structure, and in my opinion also the discussion of the results obtained (which are worthy of analysis).
2) When you refer to the SCID you do not indicate the latest version, and I am afraid you have made use of an earlier version, as with the DSM.
3) I did not find a specific and detailed reference in the text regarding ethics committee approval, there being a sample population.
Good work.
Author Response
Comment 1
The proposed manuscript is well structured in all its parts, effective and clear in its representation, the references are relevant, and there is linearity between findings, discussions, and conclusions.
Response
Thank you for this positive feedback.
Comment 2
The entire study refers to version IV of the DSM (2000), including the indication of axis subdivisions, when in fact that version has been superseded and today we use the rectified version V (5-TR, 2022). This structural deficiency invalidates the entire editorial structure, and in my opinion also the discussion of the results obtained (which are worthy of analysis).When you refer to the SCID you do not indicate the latest version, and I am afraid you have made use of an earlier version, as with the DSM.
Response
Thank you for this comments. Our data were collected before the publication of the DSM 5. We therefore, refer to DSM IV in the study, and indeed have used the DSM IV version of the SCID when we collected the data. However, since the categorical basis for the diagnostic system has not changed in this recent version, nor has the core criteria for major mood and major anxiety disorders or for the diagnosis of personality disorders, we believe that our findings can be extrapolated to the current version. We have added this important clarification to our Limitations section in the Discussion.
Comment 3
I did not find a specific and detailed reference in the text regarding ethics committee approval, there being a sample population.
Response
Please see Methods section (lines 141-143), which details the Ethics committee approvals for this study. As we detail in the manuscript, Institutional Ethics Committees at each participating clinic approved all study components and data collection adhered to acknowledged human subject protocols. Prior to study participation, participating patients and clinicians provided written informed consent.
Round 2
Reviewer 2 Report (New Reviewer)
Comments and Suggestions for Authors
Dear authors, I have read your comments and unfortunately, my line of thinking is profoundly different. I understand that the data were collected before the publication of DSM 5, but the modification of the new edition is structural, both for the part of the axes and the individual pathological particles; moreover, the current edition is the V-TR; it is your scientific duty to update and adapt your study to the most recent forms shared in the literature. Medesimo speech concerns the latest updated form; it cannot be sufficient, in my opinion, to point it out in the limits and discussions, because the error is structural and impacts the whole study design and procedure. Finally, in the text, you mentioned the reference to the ethics committee approval (and this was clear before) but you did not mention the reference number of the approval of your practice (numeric code that we need to verify the actual existence of the practice). For these reasons, I consider providing my opinion in "major revisions," as the design is too fragile and out of date, and you have not yet provided actual evidence of ethics committee approval. Good work.
Author Response
Please see the attachment. Thank you.

This manuscript is a resubmission of an earlier submission. The following is a list of the peer review reports and author responses from that submission.
Round 1
Reviewer 1 Report
Comments and Suggestions for Authors
With regard to the data collection and information used, it should be noted that the approach of bringing together information from different sources using triangulation is highly desirable. Also, the instruments chosen have good psychometric properties as indicated in the paper.
The limitations due to the type of sample used are well explained, to which it is perhaps worth remembering in the conclusions that the absence of methodological or experimental control systems in this type of study does not allow causal relationships to be established and therefore the conclusions are limited.
I would like to suggest to present table 1 divided in at least 2 tables for a better reading.
Table 2 generates certain doubts for its interpretation. It is necessary to explain the composition of the categories shown. The n in each group lead to some confusion.
The degree of significance obtained for each correlation should be shown.
As a suggestion to further enrich the conclusions, it would be desirable to consider a multivariate predictive analysis, using regression models. This would at the same time make it possible to detect relevant interactions and to analyse the role that some variables may play as effect modifiers.
The possible intervention of such modifying variables would imply that the association indices between the variables under analysis should be calculated for the different strata marking the levels of such modifying variables (e.g. by age strata).
Also, statistical control for possible confounding variables can be resolved with techniques that provide the corrected association index (e.g. odds ratio calculated with the Mantel and Haensel test).
Reviewer 2 Report
Comments and Suggestions for Authors
The manuscript “Can I leave the past behind? Associations between childhood abuse and adult psychopathology” addresses a topic pertinent to research and has implications for professional practice in relation to this target group.
The introduction provides sufficient background and includes relevant references. However, it is advisable to add pertinent bibliographical references to support the statements made in lines 50 to 56.
The research design is appropriate, and the methods are adequately described. Nonetheless, it is necessary to review the description of the inclusion criteria. On line 136, the mean and standard deviation of the months of treatment are mentioned, and the integration of this data into the article should be reviewed.
The authors have reflected on the limitations of the study. Additionally, the conclusions are clearly presented and are well-supported by the results.
